# Exploring perceptions of and attitudes towards tanning with school children, parents/carers and educators in Wales: A mixed methods study protocol for the SunChat study

**Gisselle Tur Porres**[1], **Kirsty Lanyon**[2], **Rachel Abbott**[3], **Helen Lewis**[1], **Emily Marchant**[1], **Julie Peconi**[2]*

1 Faculty of Humanities and Social Sciences, Department of Education and Childhood Studies, School of Social Sciences, Swansea University, Swansea, Wales, United Kingdom, 2 Faculty of Medicine, Swansea Trials Unit, Health and Life Science, Swansea University, Swansea, Wales, United Kingdom, 3 Cardiff and Vale University Health Board and Skin Cancer Prevention Lead for Wales, Cardiff, United Kingdom

* j.peconi@swansea.ac.uk

**Data Availability Statement:** Deidentified research data will be made publicly available when the study is completed and published.

## Abstract

### Background

Skin cancer comprises half of all cancers in England and Wales. Most skin cancers can be prevented with safer sun exposure. As over exposure as a child can greatly increase future skin cancer risk, early and accessible sun safety education and promotion of sun safe behaviours is critical. Scientists agree there is no such thing as a 'safe tan', yet the public, including children, often have positive perceptions of tanned skin. To protect against future skin cancer, it is important to understand and address these misconceptions. The Curriculum for Wales with its area for Health and Well-being, and autonomy for schools in designing curriculum content, presents an ideal way to facilitate this exploration.

### Aims

- Gather data regarding perceptions towards tanning to explore the perceived effects of a tan on health.

- Inform the development and testing of an educational toolkit for integration within the Curriculum for Wales to encourage positive health behaviours and attitudes of school children towards tanning and sun exposure.

### Methods

SunChat is a mixed methods exploratory study comprising three work streams:

1. Workshops with school children to understand their perceptions on tanning.

**Funding:** JP and GTP received funding to conduct this study from the Morgan Advanced Studies Institute (MASI) - https://www.swansea.ac.uk/masi/. The funder did not playany role in the study design, data collection and analysis, decision to publish, or preparation of the manuscript.

**Competing interests:** The authors have declared that no competing interests exist.

2. An online multiple-choice survey with parents/carers to understand perceptions, attitudes and behaviours towards tanning both for themselves and their children.

3. An informal focus group with primary school educators to explore challenges in engaging with the school community around the Health and Well-being Area in the Curriculum for Wales.

## Discussion

To date, there has been no work in Wales exploring children's, parents/carers', and educators' perceptions of tanning and how healthier attitudes can be encouraged. This study will engage with participants to scope current perceptions on tanning and the perceived effects tanning has on health. Findings will feed into future toolkit and curriculum development for health in schools in Wales and beyond.

## Introduction

### Background

Skin cancer, including melanoma, and non-melanoma (keratinocyte) also known as basal cell cancer (BCC) and squamous cell cancer (SCC), now comprises half of all cancers in England and Wales [1]. In Wales, the age-standardised incidence of keratinocyte cancer has increased by 7.1% between 2016 and 2019 –and the rate is the highest of all the UK nations [2]. Yet 86% of melanomas, 50–90% of BCCs and 50–70% of cutaneous SCCs can be prevented with safer sun habits [3]. Over exposure to the sun as a child can greatly increase future skin cancer risk, and with childhood being a critical time for forming lifelong health behaviours such as sun safety, establishing positive attitudes and behaviours to sun exposure during childhood can have important long-term implications for attitudes and behaviours throughout life [4].

These sun safe behaviours also need to address sun tanning as scientists agree that there is no such thing as a 'safe tan'. However, the public [5], university students [6], parents [7], and children [8] generally have positive perceptions of tanned skin. In Wales, a population-based online survey reported that 43.6% of respondents agreed with the statement that 'having a suntan makes me feel healthier' and this increased to 62.8% in the youngest age-group (<25 years). In the same survey, 44.5% of participants agreed with the statement that 'having a tan makes me look more attractive' [5]. These findings were also echoed in a 2017 omnibus survey reporting that 1 in 45 adults had used sunbeds in Wales and that 42% did so 'to look better' and 25% did so 'to feel healthier' [3]. With the World Health Organisation classifying exposure to sunbeds as carcinogenic to humans [9], and the use of sunbeds linked with the desire for a tan, work is urgently needed to understand and address the misconceptions of the perceived health and appearance benefits of tanning.

To date, there has been no work in Wales exploring children and their parents'/carers' and educators' perceptions of tanning and how healthier attitudes can be encouraged and adopted from a young age. The Curriculum for Wales [10] places new statutory focus on health with one of four overarching purposes to develop "healthy, confident individuals", and a designated area of learning and experience for Health and Well-being. A fundamental component of the Curriculum for Wales is the autonomy offered to schools in designing school-level curriculum

content aligned to learners' needs. This presents an ideal way to facilitate further exploration of attitudes to tanning and opportunities for the provision of curriculum materials relating to the topic of tanning and sun exposure.

Indeed, this overarching purpose of the Curriculum for Wales and the vision that all children and young people will become healthy, confident individuals aims to enable learners to know how to find information and support to keep themselves safe and well. Focusing on understanding how different health dimensions, e.g., physical, mental, emotional, and social well-being play a role in children's lives, the Health and Well-being Curriculum area also supports children to incorporate healthy attitudes and behaviours, which can include improving education and behaviours around sun safety and challenging current perceptions around tanning. To address some of these challenges and opportunities in Wales, we have designed the SunChat study, a mixed methods exploratory study.

## Study aims

- Gather data from primary school-aged children, parents/carers and primary school educators regarding perceptions of tanning and explore the perceived effects on health.

- Inform the development and testing of an educational toolkit for integration within the Curriculum for Wales to encourage positive health behaviours and attitudes of school children towards tanning and sun exposure.

## Study objectives

- To explore challenges that primary school educators face in engaging with the school community around the Health and Well-being Area of Learning and Experience in the Curriculum for Wales, specifically about healthy attitudes to tanning.

- To gather viewpoints on best ways of engaging with school-aged children (5–8 years of age) and their parents/carers about health, specifically in relation to sun tanning and sun exposure.

- To understand perceptions, attitudes and behaviours of parents/carers of primary school children in Wales regarding tanning, both for themselves and their children.

- To understand perceptions, attitudes and behaviours towards tanning and the perceived effects on health of school-aged children (5–8 years of age) in primary schools in Wales.

- Consolidate evidence to support the development of an educational toolkit for integration within the Curriculum for Wales.

## Methods

SunChat is a mixed-methods exploratory study comprising 3 work streams:

1. Qualitative workshops (n = 3) with groups of 5–10 children aged 5 to 8 years of age in Healthy School Clubs, to understand their perceptions of tanning and the perceived effects on health.

2. Online survey with parents/carers of primary school children to understand perceptions, attitudes and behaviours regarding tanning, both for themselves and their children.

3. A qualitative informal focus group with primary school educators to explore tanning perceptions/behaviours and challenges in engaging with the school community around the Health and Well-being Area of Learning and Experience in the Curriculum for Wales.

Using existing networks and school partners, we will invite three primary schools in South Wales to participate in the study as a case study school. We will choose schools who have established 'Healthy Schools Clubs', typically clubs consisting of a range of pupils and year groups whose job it is to promote healthy behaviours within the school. Overall consent will be sought from each of the school's headteacher, and each school will be invited to participate in all three work streams.

We received ethical approval to conduct the study from Swansea University's Medical School Research Sub-Committee (Ref 2022–0089). Study team members who will be directly involved with data collection in schools have up-to-date Disclosure and Barring Service (DBS) and valid safeguarding training allowing them to work with children.

As each work stream is distinct, we elaborate on methods, consent procedures, data collection and analysis separately below.

## 1. Workshops with Children

The objective of the workshops is to understand perceptions, attitudes and behaviours towards tanning and the perceived effects on health of school-aged children (5–8 years of age) in primary schools in Wales.

**Workshop informed consent and ethical considerations.** Two researchers (one facilitator and one observer) will conduct the workshops with children in each case study school's Healthy Schools Club at a time convenient to the school. The headteacher of each school will send out an information sheet and consent form to each parent/carer of a Healthy Schools Club child informing them of the planned workshop. Parents will be given 7 days to opt their child out. If a parent/carer does not opt out, that child will not be invited to participate in the workshop [11]. Workshops will be conducted in a different room to children's regular classrooms, so that children who do not have parental/carer consent to participate, will not feel excluded.

For those whose parents' consent, child-friendly information and consent sheets will be handed out before the workshop commences. We will approach the workshops from a children's right perspective [12], talking to children as empowered participants able to make decisions about taking part via ongoing, negotiated assent [11]. We will talk children through the informed consent process in an understandable and appropriate way and verbally confirm that they will be audio-recorded prior to beginning the workshop activities. Children will be informed of all the necessary workshop details and requirements so that they understand what will happen and that there are no right or wrong answers as everyone's knowledge and behaviour is different. School breaks will be adhered to and factored into the workshop planning. Additionally, children will be told they can have comfort and concentration breaks whenever needed. Once the child has consented to take part researchers will continuously check that children are comfortable through reading their body language. If any children look uncomfortable, they will be asked, using appropriate child-friendly language, if they are fine to continue and reminded that they can withdraw from the study at any time without giving an explanation.

**Workshop design.** Appropriate research design and materials for the purpose of collecting data with children is fundamental. Children's ways of expressing their views and opinions are diverse, and creative/artistic activities facilitate communication with children that expands on spoken language [13]. In each workshop, we have designed 3 activities that comprise

colouring, role-play, drawing, collage techniques, videos and posters. Each activity aims to encourage meaningful conversations with children to enhance their voices and to ensure our research aims and objectives are met.

In each workshop, we will conduct the following activities:

**Activity 1** is designed to provide an insight into how children perceive sun tanning. We will present children with an outline shape of a child on paper that will be referred to as 'a little friend'. Children will be given 'skin-colour' crayons that encompass all skin colour shades inclusively and we will ask participants to colour their 'little friend' in. We will then read a short script to children and will invite them to think of this 'little friend' after playing outdoors or going on holiday. Children will be given a new blank 'little friend' to colour in now that the 'friend' has just returned from holidays or being outdoors. The researchers will assist children if needed, engaging them in conversations and prompting them to explore children's perceptions to identify if the skin-colour of the 'little friend' has changed after being outdoors or on holidays, and why. Prompt questions will explore their thoughts about perceived beauty standards associated with being tanned and if the children relate this to healthy habits. The prompt questions below will be used during the activity:

- *How our little friend looks before playing on the beach, the park. . ., on a summer day*?

- *And how they look after, when they are back home, indoors*?

- *Is there any difference*? *Do they look the same*?

- *Why did you choose this colour (same or different)*?

- *Is your little friend tanned*?

- *Do they look pretty*? *(Show them the first figure; and then the second figure) Depending on answers, why do they look pretty*?

- *Who looks healthier*? *(Show them the first figure; and then the second figure) Depending on answers, why do they look healthier*?

- *Does your little friend (dis-)like to be tanned*? *Do you know why*?

- *Is there anything you (dis-)like about tanning*? *What*? *Why*?

Researchers will use inclusive language that embraces ethnic diversity to discuss skin colour after tanning. All questions that researchers will ask associated with perceived beauty standards will solely be oriented to being 'tanned' or 'untanned'. Please see S1 Appendix for an outline of this Activity.

**Activity 2** will gather data to understand what healthy habits children already know about how to stay safe in the sun. In this activity we will use different materials laid out centrally on a classroom table for children to access easily and will invite them to draw a picture of themselves doing their favourite activities in the sun on their own or with their family, friends or pets. Children can also choose images and cut-outs from magazines with a range of items or clothing related to sun safety to glue on their drawing (collage). The researcher will also assist children to set the scene, prompting them with questions such as "What do they love doing in the sun?" and "What do they need to enjoy the sun safely?". Children will then be invited to watch a short video [14] showing a story about sun safety habits and behaviours such as wearing a hat and putting on suncream. They will then be given time to consider if they would like to add something to their drawings after watching the video to protect themselves, friends, pets, and their families from the sun. Please see S2 Appendix for an outline of this Activity.

**Activity 3** is designed to meet the overall aim of encouraging healthy behaviours and to promote sun safety which will help to inform the development the educational toolkit. In this activity, children will be helping other children to promote sun safety and healthy habits and learn why it is healthier not to tan on purpose. We will invite children to work in groups to draw, use images and cut-outs or write down things that are healthy/safe and things that are unhealthy/unsafe to do in the sun. These images will then be used to create a large poster. We will guide children in a group conversation related to healthy and unhealthy habits; sun safety and unsafe behaviours and will let them decide what goes in one of two columns on the poster under the headings "YES" (healthy/safe) or "NO" (unhealthy/unsafe),. Researchers will again assist children and facilitate discussions with them around healthy/unhealthy habits and suggesting some of the behaviours (i.e. wear a hat) shown in the video [14] in Activity 2. During the activity, researchers will encourage children to think about their own sun safety and healthy behaviours and what they believe is a healthy approach to sun safety. Please see S3 Appendix for an outline of this Activity.

**Workshop data collection, management and analysis.**    All workshop data will be audio recorded for later transcription by a secure third-party transcription service. Any names mentioned in the audio recordings will be redacted from the final anonymised transcripts. The observer will record detailed field notes regarding children's perceptions to tanning and the relation to health benefits and interactions amongst participants. Pictures of resources developed in the activities with children will be taken and subsequently anonymised before used in the data analysis.

Data collected on paper and completed workshop resources will be kept in a locked filing cabinet with restricted access to approved members of the research team only. Electronic data and scanned copies will be held securely on Swansea University servers which are subject to physical, electronic, and managerial procedures to safeguard and secure the information collected.

We will explore differences between perceptions, attitudes and reported behaviours towards tanning to identify whether there are any differences between knowledge and healthy practices from each of the activities. We will use NVivo12, an online qualitative data analysis software package to analyse workshop transcripts and children's outputs using thematic analysis following the recommended 6 step process: familiarisation; coding; generating themes; reviewing themes; defining and naming themes and writing up [15]. We will analyse the resources and pictures children create during the workshop using content analysis. [16].

Archiving will be conducted following Swansea Trials Unit Standard Operating Procedures (SOPs). All data will be held in archive for a period of 5 years following the completion of the project.

## 2. Online parent/carer survey

We have designed a short online multiple choice survey to help understand current perceptions, attitudes and behaviours of parents/carers of primary school children in Wales regarding tanning, both for themselves and their children.

**Survey informed consent and ethical considerations.**    Parents/carers must have at least one child of primary school age to participate. Parent/carers who access the survey will be introduced to its purpose in a short introduction text and directed to an online participant information sheet with full details about the survey and data protection which they are able to download and read in their own time. If the parent/carer is happy to continue, they will progress through the survey using the clear online navigation buttons. Due to the low-risk nature of participation, consent for survey completion will be assumed if the parent/carer completes

the survey and the response is submitted. Respondents will be instructed that they do not have to complete the survey and can close out the survey at any time if they wish to withdraw and their answers will not be recorded. There will be an estimated time for completion provided within the survey introduction text along with contact details of key researchers on the study for any questions or to raise concerns.

**Survey design and distribution.** We scoped the current literature and collated various questions about beliefs and perceptions of tanning from similar surveys available from published sun safety research [17–20] and adapted/included questions to include in an online survey.

We will collect basic demographic data such as location, job role and gender of survey respondents. Respondents will also be asked the age of their eldest child in primary school and their child's year group and asked to indicate their self-assessed skin colour using the Fitzpatrick skin colour scale [21]. The survey consists of Likert scale type questions (strongly agree to strongly disagree) related to how healthy and/or attractive respondents feel a tan looks on themselves and their child along, with how positively/negatively they feel their child thinks about tanning. For example, questions include "I feel healthier with a suntan", "my child thinks they look better with a tan", and "my child feels pressure from social media, tv and video streaming platforms to have a tan". Please see S4 Appendix for a copy of our survey.

The number of survey respondents will not be pre-determined, and due to the nature of the study design, no specific sample size is set. We will aim to receive as many responses as possible. Survey advertisements will be created and ethically approved to help disseminate the survey links. These will contain QR codes which can be scanned using an eligible smartphone for direct access to the survey. All survey materials including the information sheets, advertisements and the survey itself will be available in both English and Welsh.

The survey links and QR codes will be distributed using several pathways in an attempt to reach as wide a variety of parents/carers as possible including a mix of genders, ethnicities, ages, number of children and professions. Firstly, we will distribute the survey to our case study schools to disseminate to parents of their enrolled pupils. We will then look to promote the survey publicly using other methods such as promotion on social media and University email lists. Survey advertisements will also be uploaded to public websites including the Swansea University website. We will also ask current contacts in Health and Care Research Wales, the charity, Tenovus Cancer Care and our Public Involvement Representative, to promote the survey link on their social media pages. Finally, we will post the link on relevant parent and carer online forums and encourage others to share the link. We will record all survey promotion activities on a designated bespoke survey launch plan designed for the study.

Eligibility will be self-reported by the parent/carers using a screening question asking if the respondent has at least one child of primary school age. Anyone who responds "no" to this question will not be able to continue. Parents who have multiple children of primary school age will be asked to consider their responses thinking about their eldest child only. Although we will aim to capture respondents from Wales, public dissemination of the link will mean that anyone able to click on the link will be able to complete the survey, so the respondent will be asked to select their location from a pre-defined list.

Again, we will aim to receive as many survey responses as possible and will not enforce an upper limit on the number of survey responses we will accept.

**Survey data collection, management and analysis.** Survey data will be entered directly onto the JISC Online Survey platform by respondents and will be held centrally throughout active data collection. JISC is a highly secure survey platform which is a GDPR compliant, online survey tool designed for academic research, education, and public sector organisations, certified to ISO 27001. JISC also provides daily back-ups for all data collected within online

surveys. All user accounts on JISC Online Surveys are password protected and unique to the individual.

We will analyse data in an accepted statistical software package using simple statistics, such as measures of distribution means, cross tabulations, and tests of association such as t-tests for continuous variable and Pearson's chi square for categorical variables. We will analyse free text responses using thematic analysis in NVivo following the recommended 6 step process: familiarisation; coding; generating themes; reviewing themes; defining and naming themes and writing up [15]. Two researchers who have qualitative analysis experience will conduct the thematic analysis.

Survey data will be archived following Swansea Trials Unit SOPs. All data will be held in archive for a period of 5 years following the completion of the project.

## 3. Primary school educator focus group

The objective of the focus group is to explore the challenges that primary educators face in engaging with the school community around the Health and Well-being Area of Learning and Experience in the Curriculum for Wales, specifically about healthy attitudes to tanning. We will gather viewpoints on the best ways of engaging with school-aged children (5–8 years of age) and their parents/carers about health, specifically in relation to sun tanning and sun exposure.

**Focus group informed consent and ethical considerations.**   Head teachers from our case study schools will share information about the focus group to their school educators who will be invited to submit an expression of interest to take part. Researchers will follow up with potential participants to provide information about the consent process and to receive written consent from participants. Focus groups will be conducted at a suitable time to maximise availability options for participants, with consideration of school schedules and hours. Participants will be given the opportunity to ask any questions before beginning the focus group and reminded that they will be audio recorded for transcription purposes. Researchers will also remind all participating members how long the discussion is likely to take, that they are not obliged to answer any questions they do not wish to, and that participation is optional so they may withdraw at any time without giving an explanation. Facilitators will also offer comfort breaks if or when they are needed.

**Focus group design.**   The focus group will be conducted online using video conferencing software. We will create a semi-structured topic guide with discussion points on opinions about providing tanning education for primary school children and what the current awareness is about tanning in their schools. We will also cover opinions about how to best promote the message of safer sun exposure and tanning and if there would be any limitations or challenges delivering this awareness in primary school children. Researchers will conduct the focus group in line with focus group facilitation guidance, which will be given to participants before commencing any discussions to ensure that everyone has an opportunity to contribute equally and there is no overtalking. Two focus group facilitators will be present to guide and encourage discussions and to ensure that participants feel comfortable sharing their opinions and current practices in their schools. We will offer consented participants the choice of conducting the focus group in English or Welsh.

**Focus group data collection, management and analysis.**   We will audio record the focus group with participant consent and will have the recordings transcribed by a secure third-party transcription service, without identifying participants. The transcripts will be checked and confirmed as accurate prior to analysis. Identifying data, including informed consent forms and audio recordings, will be held electronically in a secure folder with restricted access

on Swansea University servers which are subject to physical, electronic, and managerial procedures to safeguard and secure the information collected.

We will analyse the data to identify key themes around tanning, and challenges around the Health and Well-being Area of Learning and Experience for primary school educators. Again, we will use NVivo to handle qualitative analysis and will perform thematic analysis following the recommended 6 step process: familiarisation; coding; generating themes; reviewing themes; defining and naming themes and writing up [15].

As with all data collected, archiving will be conducted following Swansea Trials Unit SOPs. All data will be held in archive for a period of 5 years following the completion of the project.

## Impact and dissemination

To enable dissemination of study results in an accessible way which can be understood by children, a collaborative infographic poster will be designed to showcase schools' and children's involvement in the workshops. Children will be informed that the poster will be used publicly and that they will not be identified from any poster content. Survey results will also be collated into a lay-person friendly format, highlighting key findings. Results from this collaborative study will also be used to inform development and testing of an educational toolkit to encourage children's healthy behaviours towards tanning and sun exposure as part of future work. Further findings from all workstreams will be written up for publications in suitable journals.

We will design a SunChat webpage hosted on a Swansea University website to promote real-time study news and provide public updates of the study progress. Video logs (vlogs) will also be created and uploaded online to provide lay-friendly updates and public insights into study activities over the study lifetime.

## Patient and public involvement

Parent/carer involvement in all aspects of the study design will be considered crucial for aligning with study aims in an appropriate way, the success of data collection and maximising the resulting impact of the study. We will be guided by the UK standards for public involvement [22] and endeavour to meet the standards at every opportunity presented while collaborating with public representatives.

Drafts of the study online survey will be reviewed by a parent representative who has child (ren) of a primary school age for appropriateness/relevance of survey questions, length of time to complete and layout/design choices. Our representative will also help to test the final survey against criteria such as user acceptance, screen testing and confirming the survey works as expected. Training which is determined as necessary to conduct any study tasks which may fall outside of the current skills of our representative will be provided by members of the project team, including but not limited to introductory project meetings, ongoing support with project team requests and protocol training. All feedback will be considered and incorporated where feasible into the survey, and updates will be made to the final version using parent/carer feedback. Our parent/carer representative will also aid in circulating the survey link to their peers, their children's school and social media and provide invaluable advice about how to best reach out to parents/carers with our survey. Our parent/carer representative will be appropriately reimbursed according to current recommendations for public involvement payments for their time and any expenses incurred as part of their involvement. Finally, we will look to include our representative in study dissemination activities, such as forwarding on publications and lay friendly study outputs to their social groups and seeking advice for key findings they determine as important to highlight in results summaries.

## Discussion

With skin cancer an issue in Wales and beyond, rising rates will not only continue to effect health and well-being but will overburden NHS dermatology services and care for other skin conditions, affecting future use of services for today's children. As suggested by an Australian study [23], the implementation of public health prevention programmes show relevant health and economic cost benefits to reduce skin cancer burden. Developing and embedding positive health-related behaviours should begin in childhood, as these years give the ideal opportunity for children to develop prevention strategies [24]. One study [4] suggests that prevention strategies and healthy behaviours implemented during teenage years may be too late and not as effective due to tanning being considered 'attractive' by adolescents, so recommend that it is more beneficial to encourage healthier behaviours during the formative years of life.

### Study strengths

This study builds on the innovative and holistic education processes proposed in the Curriculum for Wales with a statutory focus on Health and Well-being to address the community impact of a public health issue in Wales. Here, we take the known problem of skin cancer and address it in a novel way by engaging directly with children through creative activities identified as best practice in facilitating and prioritising their participation [25]. Also, creative and artistic approaches are known as young children's preferred ways of communicating their views, thoughts and opinions [26]. As academics that do research with school-aged children, it is our responsibility to explore children's perceptions in ways that facilitate their open communication, and ensuring their voices are heard. Our proposed workshops at schools will include activities that use child-friendly methodologies and appropriate language for collecting information and ensuring that we accurately capture, analyse and address children's perceptions to tanning.

Overall, workshop activities will support the development of a toolkit that can include proposed lessons' plans with resources and allow teachers to integrate it into their curriculum planning. Health and Wellbeing is an area of learning and experience of Curriculum for Wales that aims at providing opportunities for children to understand how health and wellbeing are interconnected. It also supports the purpose of becoming informed and ethical citizens that can manage risks, understand, and develop healthy attitudes, e.g. towards sun safety. Talking about attitudes, behaviours, and perceptions of tanning can also lead to assembly discussions about health risks and how to stay safe in the sun. In this regard, conversations about becoming responsible for one's own behaviour and taking care of our loved ones to be safe in the sun, could also support the purpose of becoming informed and ethical citizens. Exploratory activities, e.g., Posters developed with children in Activity 3 can also inform the toolkit and be used in assemblies with children's consent, with a suggested topic to discuss healthy and unhealthy sun safe behaviours.

As well as exploring perceptions of tanning and encouraging conversations towards healthy attitudes with children, school educators, and families, this study will also create opportunities for continued inter-disciplinary collaboration between health and education academics. The study has the potential to lead into further research where we would look to evaluate our toolkit and future curriculum development for schools on questioning perceptions of tanning and the effects on young children's health and well-being, aligned to the views of children themselves. Taking responsibility for health-related behaviours including attitudes to tanning should begin in childhood, and hence is the ideal opportunity for children to develop prevention strategies.

## Study limitations

Due to time and budget constraints, we are limited to inviting case study schools in the Swansea area in South Wales. Further representation is needed from more primary schools in Wales from varying locations and demographic backgrounds to provide a clearer picture of the situation around tanning beliefs in children and their parents/carers/educators. Researchers on the study team will incorporate this requirement into the design of a potential larger-scale study in the future to include participants from across the UK to ensuring representation from demographically and ethnically diverse populations. However, the online survey for this study will be distributed widely to all parents in Wales using several avenues for dissemination and we therefore hope to receive responses from a diverse range of parents/carers.

We hope that by sharing this protocol, we can help raise awareness of the importance of challenging and changing the prevailing assumption that having a tan is desirable and support schools in empowering and educating children to become healthy and informed adults.

## Supporting information

**S1 Appendix. SunChat Workshop Activity 1.**
(DOCX)

**S2 Appendix. SunChat Workshop Activity 2.**
(DOCX)

**S3 Appendix. SunChat Workshop Activity 3.**
(DOCX)

**S4 Appendix. SunChat Online Survey v1.1_22_03_2023.**
(DOCX)

## Acknowledgments

We would like to acknowledge all participants and collaborators including the children, parents/carers and primary school teachers who will provide the data needed for this research giving up their valued time to contribute to the study.

## Author Contributions

**Conceptualization:** Gisselle Tur Porres, Rachel Abbott, Helen Lewis, Emily Marchant, Julie Peconi.

**Data curation:** Kirsty Lanyon.

**Funding acquisition:** Gisselle Tur Porres, Julie Peconi.

**Investigation:** Gisselle Tur Porres, Kirsty Lanyon, Emily Marchant.

**Methodology:** Gisselle Tur Porres, Rachel Abbott, Helen Lewis, Emily Marchant, Julie Peconi.

**Project administration:** Kirsty Lanyon.

**Resources:** Gisselle Tur Porres.

**Supervision:** Gisselle Tur Porres, Rachel Abbott, Julie Peconi.

**Validation:** Kirsty Lanyon.

**Writing – original draft:** Gisselle Tur Porres.

**Writing – review & editing:** Kirsty Lanyon, Rachel Abbott, Helen Lewis, Emily Marchant, Julie Peconi.

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
