## [Decision Letter · Decision Letter 0]

12 Feb 2024

PONE-D-23-38933Exploring perceptions of and attitudes towards tanning with school children, parents/carers and educators in Wales: A mixed methods study protocol for the SunChat study.PLOS ONE

Dear Dr. Peconi,

Thank you for submitting your manuscript to PLOS ONE. After careful consideration, we feel that it has merit but does not fully meet PLOS ONE’s publication criteria as it currently stands. Therefore, we invite you to submit a revised version of the manuscript that addresses the points raised during the review process.

We look forward to receiving your revised manuscript.

Kind regards,

Annesha Sil, PhD

Associate Editor

PLOS ONE

Journal Requirements:

**Additional Editor Comments:** The manuscript has been evaluated by 2 reviewers, and their comments are available below.

The reviewers have raised a few concerns. They feel the manuscript would benefit from a more thorough description of the toolkit proposed, more details about the measures and items to be evaluated in the surveys and methodological details such as sample sizes for the online surveys. Additionally, as per PLOS One's guidelines for study protocols and data availability, kindly indicate where/how data would be available for this protocol. Could you please carefully revise the manuscript to address all comments raised?

Reviewers' comments:

Reviewer's Responses to Questions

**Comments to the Author**

1. Does the manuscript provide a valid rationale for the proposed study, with clearly identified and justified research questions?

Reviewer #1: Yes

Reviewer #2: Partly

2. Is the protocol technically sound and planned in a manner that will lead to a meaningful outcome and allow testing the stated hypotheses?

Reviewer #1: Yes

Reviewer #2: Yes

3. Is the methodology feasible and described in sufficient detail to allow the work to be replicable?

Reviewer #1: Yes

Reviewer #2: Yes

4. Have the authors described where all data underlying the findings will be made available when the study is complete?

Reviewer #1: Yes

Reviewer #2: Yes

5. Is the manuscript presented in an intelligible fashion and written in standard English?

Reviewer #1: Yes

Reviewer #2: Yes

6. Review Comments to the Author

You may also provide optional suggestions and comments to authors that they might find helpful in planning their study.

Reviewer #1: Thank you to the author/s. It is recommended to review the text in terms of grammar. Results of the word repetitions should be avoided and given in a shorter, understandableway.

Reviewer #2: The aim of this effort and its plan to work via the Healthy Schools clubs are very worthwhile. It is essential to help develop constructive attitudes, beliefs, and habits related to health early in life. The description of study procedures is well organized, easy to follow, and a pleasure to read. The methods for working with young children are highly creative, well planned, and thoughtfully child-centered. Data collection, transcription, and analysis are described in careful detail. The overall methodology of the study is feasible and provided enough detail for replication. With just a bit more detail in other areas, I believe the full picture of the protocol could be clarified. I suggest the following be addressed for completeness and clarity:

a) An aim of the study is the development of a “pilot toolkit for integration within the Curriculum for Wales.” If more specifics are known, I would be curious to know what a toolkit includes and how it would be integrated and implemented within the Health and Well-being component of the Curriculum.

b) For someone unfamiliar with sun protection, examples of sun safety habits and behaviours (especially those that can be adopted by children and families) would be informative.

c) It would strengthen the paper to provide an example of how the information gleaned from the exploratory activities with children could inform the toolkit content.

d) Measurement is a key part of a study protocol. The study procedures would be more complete with a description of the measures to be used or examples of survey items, especially those related to perceptions of and attitudes toward tanning.

e) A sample size is included for the workshops with children. A sample of 15-30 children seems adequate. Estimated or targeted sample sizes should also be included for the online survey of parents/carers and the educator focus group.

f) Parent skin colour is not always a proxy for the child. The authors may want to consider asking the children to indicate their self-assessed skin colour using the Fitzpatrick skin colour scale or a simplified version of it. In addition, or alternatively, the research er could have an observer code the observed skin colour of each child, if this data would be valuable.

g) It was not clear whether the authors registered on a research platform that is appropriate for the study type.

7. PLOS authors have the option to publish the peer review history of their article (what does this mean?). If published, this will include your full peer review and any attached files.

Reviewer #1: **Yes: **Adem SÜMEN

Reviewer #2: **Yes: **Mary Klein Buller

---

## [Author Response · Author response to Decision Letter 0]

25 Mar 2024

Additional Editor Comments:

1) The manuscript has been evaluated by 2 reviewers, and their comments are available below.

The reviewers have raised a few concerns. They feel the manuscript would benefit from a more thorough description of the toolkit proposed, more details about the measures and items to be evaluated in the surveys and methodological details such as sample sizes for the online surveys. 

Additionally, as per PLOS One's guidelines for study protocols and data availability, kindly indicate where/how data would be available for this protocol. Could you please carefully revise the manuscript to address all comments raised?

Our response: Thank you again for giving us the opportunity to address the reviewers’ comments. Below and using tracked changes in the manuscript, we detail how we have addressed these comments including more details about the toolkit and the measures and items to be evaluated in the survey. We also have included as supplementary files, our workshop activities and online survey. 

I’m sorry but the link to protocol data availability on your website is showing as unavailable so we couldn’t amend this sentence at this point in time. 

Reviewers' comments:

Reviewer's Responses to Questions 

Comments to the Author

1. Does the manuscript provide a valid rationale for the proposed study, with clearly identified and justified research questions?

Reviewer #1: Yes

Reviewer #2: Partly

Our Response: Thank you for noting this. 

2. Is the protocol technically sound and planned in a manner that will lead to a meaningful outcome and allow testing the stated hypotheses?

Reviewer #1: Yes

Reviewer #2: Yes

Our Response: Thank you for noting this. 

3. Is the methodology feasible and described in sufficient detail to allow the work to be replicable?

Reviewer #1: Yes

Reviewer #2: Yes

Our Response: Thank you. We have also strengthened this area based on Reviewer feedback and included more information about the methods we will use, particularly with respect to the online survey. 

4. Have the authors described where all data underlying the findings will be made available when the study is complete?

Reviewer #1: Yes

Reviewer #2: Yes

Our Response: Thank you. As this is a protocol paper, we currently have no data to share. 

5. Is the manuscript presented in an intelligible fashion and written in standard English?

Reviewer #1: Yes

Reviewer #2: Yes

Our Response: Thank you, we have also made minor edits to the text, in line with Reviewer #1’s comments below. We thank you for this comment as we feel we have now made the paper more understandable and readable. 

6. Review Comments to the Author

Reviewer #1: Thank you to the author/s. It is recommended to review the text in terms of grammar. Results of the word repetitions should be avoided and given in a shorter, understandable way.

Our Response: Thank you, we have now gone through the manuscript and made minor edits for grammar and repetition. We apologise for missing these in the first place and thank you for picking this up. As we said above, we feel the paper is now easier to read. 

Reviewer #2: The aim of this effort and its plan to work via the Healthy Schools clubs are very worthwhile. It is essential to help develop constructive attitudes, beliefs, and habits related to health early in life. The description of study procedures is well organized, easy to follow, and a pleasure to read. The methods for working with young children are highly creative, well planned, and thoughtfully child-centered. Data collection, transcription, and analysis are described in careful detail. The overall methodology of the study is feasible and provided enough detail for replication. With just a bit more detail in other areas, I believe the full picture of the protocol could be clarified. 

I suggest the following be addressed for completeness and clarity:

a) An aim of the study is the development of a “pilot toolkit for integration within the Curriculum for Wales.” If more specifics are known, I would be curious to know what a toolkit includes and how it would be integrated and implemented within the Health and Well-being component of the Curriculum.

Our Response: Thank you for this comment. We have added additional information to the manuscript to help the reader understand what a potential toolkit could include. 

In Wales, each school could develop its own curriculum towards the four purposes that support children becoming: (a) ambitious, capable learners, ready to learn throughout their lives; (b) enterprising, creative contributors, ready to play a full part in life and work; (c) ethical, informed citizens of Wales and the world; (d) healthy, confident individuals, ready to lead fulfilling lives as valued members of society. 

Overall, workshop activities will support the development of a toolkit that can suggest lesson plans with resources and allow teachers to integrate it in their curriculum planning. Health and Wellbeing is an area of learning and experience of Curriculum for Wales that aims at providing opportunities for children to understand how health and wellbeing are interconnected. It also supports the purpose of becoming informed and ethical citizens that can manage risks, understand, and develop healthy attitudes, e.g. towards sun safety. Talking about attitudes, behaviours, and perceptions of tanning can lead to assembly discussions about health risks and how to stay safe in the sun to look after ourselves and others. In this regard, conversations about becoming responsible of our own behaviours and taking care of our loved ones to be safe in the sun, could also support the purpose of becoming informed and ethical citizens. Exploratory activities, e.g., Posters developed with children in Activity 3 (see supplementary copy of activity 3 planning sheet), can inform the toolkit and be used in assemblies with children’s consent, to discuss healthy and unhealthy ways to enjoy the sun.

b) For someone unfamiliar with sun protection, examples of sun safety habits and behaviours (especially those that can be adopted by children and families) would be informative.

Our Response: We have now added in some sun protection examples to the text (e.g. wearing a hat). We have also included a reference to the short video that will show sun safety habits and behaviours to be shared with children in Activity 2 and uploaded a copy of the proposed activities as supplementary files. After watching the video, pupils will be given time to consider the messaging and add new elements to their drawings (if they wish to do so). 

c) It would strengthen the paper to provide an example of how the information gleaned from the exploratory activities with children could inform the toolkit content.

Our Response: We have addressed this comment in 6a above about how activities could inform toolkit development. 

d) Measurement is a key part of a study protocol. The study procedures would be more complete with a description of the measures to be used or examples of survey items, especially those related to perceptions of and attitudes toward tanning.

Our Response: Thank you for this comment. We have added examples of survey items related to perceptions of tanning to the manuscript. We will also upload a copy of the survey as a supplementary file to this manuscript. 

e) A sample size is included for the workshops with children. A sample of 15-30 children seems adequate. Estimated or targeted sample sizes should also be included for the online survey of parents/carers and the educator focus group.

Our Response: This is a public facing survey. The number of participants is not pre-determined, and due to the nature of the study design, no specific sample size is set. The survey is open access, and we have no way of controlling how many people see the link, or choose to share, it or complete the survey. We have designed it this way to elicit as many responses as possible in a short period of time. 

The focus group size is linked to the 3 schools we worked with, we will invite teachers from each school to participate to provide context for, and supplement findings from the workshop. 

We have updated the manuscript accordingly.

f) Parent skin colour is not always a proxy for the child. The authors may want to consider asking the children to indicate their self-assessed skin colour using the Fitzpatrick skin colour scale or a simplified version of it. In addition, or alternatively, the researcher could have an observer code the observed skin colour of each child, if this data would be valuable.

Our Response: Thank you for this suggestion. The self- assessed skin colour is a very relevant suggestion to consider in future studies, this is outside scope of the current study, but it is interesting for future research. In this study, we will consider skin colour in workshops (activity 1) where children are given skin colour crayons that encompass all skin colours shade inclusively. Researchers will use inclusive ethnic language when talking about skin colour, and it will make clear to children that we are referring to skin colour before and after tanning. 

g) It was not clear whether the authors registered on a research platform that is appropriate for the study type.

Our Response: As this is not a clinical trial, we have not registered our study protocol on a research platform. However, we have made information about the study available on Swansea Trial Unit’s website. SunChat: SUN Safety Conversations about Healthy Attitudes to Tanning - Swansea Trials Unit

7. PLOS authors have the option to publish the peer review history of their article (what does this mean?). If published, this will include your full peer review and any attached files.

Do you want your identity to be public for this peer review? For information about this choice, including consent withdrawal, please see our Privacy Policy.

Reviewer #1: Yes: Adem SÜMEN

Reviewer #2: Yes: Mary Klein Buller

Our Response: Thank you both again for your helpful comments. 

https://eur03.safelinks.protection.outlook.com/?url=https%3A%2F%2Fjournals.plos.org%2Fplosone%2Fs%2Ffile%3Fid%3DwjVg%2FPLOSOne_formatting_sample_main_body.pdf&data=05%7C02%7CJ.Peconi%40Swansea.ac.uk%7Ce2fbdd9b460146cd57be08dc2bc9aeb0%7Cbbcab52e9fbe43d6a2f39f66c43df268%7C0%7C0%7C638433392530445014%7CUnknown%7CTWFpbGZsb3d8eyJWIjoiMC4wLjAwMDAiLCJQIjoiV2luMzIiLCJBTiI6Ik1haWwiLCJXVCI6Mn0%3D%7C0%7C%7C%7C&sdata=6QtVfu0NaKDmBpdXA3FlANs8K%2Bf2cSnJVeUXln8MFpo%3D&reserved=0 and

https://eur03.safelinks.protection.outlook.com/?url=https%3A%2F%2Fjournals.plos.org%2Fplosone%2Fs%2Ffile%3Fid%3Dba62%2FPLOSOne_formatting_sample_title_authors_affiliations.pdf&data=05%7C02%7CJ.Peconi%40Swansea.ac.uk%7Ce2fbdd9b460146cd57be08dc2bc9aeb0%7Cbbcab52e9fbe43d6a2f39f66c43df268%7C0%7C0%7C638433392530451702%7CUnknown%7CTWFpbGZsb3d8eyJWIjoiMC4wLjAwMDAiLCJQIjoiV2luMzIiLCJBTiI6Ik1haWwiLCJXVCI6Mn0%3D%7C0%7C%7C%7C&sdata=k1vxiazAuF0HvM%2FK52EBixHkOD9mMpHV1X0eeiybeBI%3D&reserved=0

"We want to acknowledge the Morgan Advanced Studies Institute (MASI) at Swansea University for funding this important exploratory study. We would also like to acknowledge all participants and collaborators including the children, parents/carers and primary school teachers who will provide the data needed for this research and have given up their valued time to contribute to the study."

"JP and GTP received funding to conduct this study from the Morgan Advanced Studies Institute (MASI) - https://www.swansea.ac.uk/masi/. The funder did not play any role in the study design, data collection and analysis, decision to publish, or preparation of the manuscript."

Our response: We have now removed the reference to our funders on page 5 and also from the acknowledgement sections in our manuscript. We apologise for this oversight. 

Please can the revised funding statement read: We received funding for this study from Swansea University’s Morgan Advanced Studies Institute (MASI) - https://www.swansea.ac.uk/masi/. The funder did not play any role in the study design, data collection and analysis, decision to publish, or preparation of the manuscript."

Our response: Please see above.

3. In the online submission form, you indicated that "Deidentified research data can be made publicly available on request when the study is completed and published."

Our Response

---

## [Decision Letter · Decision Letter 1]

24 Apr 2024

Exploring perceptions of and attitudes towards tanning with school children, parents/carers and educators in Wales: A mixed methods study protocol for the SunChat study.

PONE-D-23-38933R1

Dear Dr. Peconi,

We’re pleased to inform you that your manuscript has been judged scientifically suitable for publication and will be formally accepted for publication once it meets all outstanding technical requirements.

Kind regards,

Caroline Watts, PhD

Academic Editor

PLOS ONE

Additional Editor Comments (optional):

Reviewers' comments:

Reviewer's Responses to Questions

**Comments to the Author**

1. Does the manuscript provide a valid rationale for the proposed study, with clearly identified and justified research questions?

Reviewer #1: Yes

Reviewer #2: Yes

2. Is the protocol technically sound and planned in a manner that will lead to a meaningful outcome and allow testing the stated hypotheses?

Reviewer #1: Yes

Reviewer #2: Yes

3. Is the methodology feasible and described in sufficient detail to allow the work to be replicable?

Reviewer #1: Yes

Reviewer #2: Yes

4. Have the authors described where all data underlying the findings will be made available when the study is complete?

Reviewer #1: Yes

Reviewer #2: Yes

5. Is the manuscript presented in an intelligible fashion and written in standard English?

Reviewer #1: Yes

Reviewer #2: Yes

6. Review Comments to the Author

You may also provide optional suggestions and comments to authors that they might find helpful in planning their study.

Reviewer #1: It appears that the authors made the changes specified by the reviews. The article is acceptable in its current form.

Reviewer #2: The authors' provision of additional information and explanation to enhance completeness and clarity for the readers is recognized and appreciated.

7. PLOS authors have the option to publish the peer review history of their article (what does this mean?). If published, this will include your full peer review and any attached files.

Reviewer #1: No

Reviewer #2: **Yes: **Mary Klein Buller, MA

---

## [Editor Report · Acceptance letter]

9 May 2024

PONE-D-23-38933R1 

PLOS ONE

Dear Dr. Peconi, 

I'm pleased to inform you that your manuscript has been deemed suitable for publication in PLOS ONE. Congratulations! Your manuscript is now being handed over to our production team.

Kind regards, 

on behalf of

Dr. Caroline Watts 

Academic Editor

PLOS ONE